# Role of Epithelial–Mesenchymal Plasticity in Pseudomyxoma Peritonei: Implications for Locoregional Treatments

**DOI:** 10.3390/ijms21239120

**Published:** 2020-11-30

**Authors:** Maria Luisa Calabrò, Nayana Lazzari, Giulia Rigotto, Marco Tonello, Antonio Sommariva

**Affiliations:** 1Immunology and Molecular Oncology, Veneto Institute of Oncology IOV-IRCCS, I-35128 Padua, Italy; nayana.lazzari@iov.veneto.it (N.L.); giulia.rigotto@iov.veneto.it (G.R.); 2Surgical Oncology of the Esophagus and Digestive Tract, Veneto Institute of Oncology IOV-IRCCS, I-35128 Padua, Italy; marco.tonello@iov.veneto.it; 3Advanced Surgical Oncology, Surgical Oncology of the Esophagus and Digestive Tract, Veneto Institute of Oncology IOV-IRCCS, I-35128 Padua, Italy; antonio.sommariva@iov.veneto.it

**Keywords:** pseudomyxoma peritonei, PMP, epithelial-mesenchymal transition, epithelial-mesenchymal plasticity, EMT, EMP, cytoreductive surgery, hyperthermic intraperitoneal chemotherapy, CRS/HIPEC

## Abstract

The mechanisms by which neoplastic cells disseminate from the primary tumor to metastatic sites, so-called metastatic organotropism, remain poorly understood. Epithelial–mesenchymal transition (EMT) plays a role in cancer development and progression by converting static epithelial cells into the migratory and microenvironment-interacting mesenchymal cells, and by the modulation of chemoresistance and stemness of tumor cells. Several findings highlight that pathways involved in EMT and its reverse process (mesenchymal–epithelial transition, MET), now collectively called epithelial–mesenchymal plasticity (EMP), play a role in peritoneal metastases. So far, the relevance of factors linked to EMP in a unique peritoneal malignancy such as pseudomyxoma peritonei (PMP) has not been fully elucidated. In this review, we focus on the role of epithelial–mesenchymal dynamics in the metastatic process involving mucinous neoplastic dissemination in the peritoneum. In particular, we discuss the role of expression profiles and phenotypic transitions found in PMP in light of the recent concept of EMP. A better understanding of EMP-associated mechanisms driving peritoneal metastasis will help to provide a more targeted approach for PMP patients selected for locoregional interventions involving cytoreductive surgery and hyperthermic intraperitoneal chemotherapy.

## 1. Introduction

The peritoneal cavity is a well-known metastatic site for several malignancies. The mechanism of dissemination of tumors into the peritoneum represents a peculiar metastatic route distinct from lymphatic and hematogenous ones, as cancer cells detach from the primary tumor, disseminate inside the peritoneal cavity and eventually implant on the peritoneal submesothelium [1,2,3].

Pseudomyxoma peritonei (PMP) is an anatomo-clinical condition most commonly secondary to peritoneal metastases from a perforated mucinous appendiceal tumor. The clinical picture is determined by implantation of neoplastic cells onto peritoneal surfaces with the production of a large amount of intraperitoneal mucin (MUC). Since the introduction in the early nineties of cytoreductive surgery (CRS) associated with hyperthermic intraperitoneal chemotherapy (HIPEC) as locoregional treatment, PMP represents one of the most established indications for this treatment, and, critically, the only one with the potential to achieve cure or long-term disease control [4,5]. Given the rarity of this disease, the role of other therapeutic strategies remains unproven and difficult to verify within properly designed studies. Few data are available on the efficacy of systemic treatments, i.e., chemotherapy or targeted therapy, but, in general, a relative unresponsiveness is reported. For these reasons, the identification of potential molecular targets that could contribute to improved patient stratification is needed to extend the therapeutic opportunities of these patients. Although immortalized cell lines and animal PMP models have recently been developed, the molecular mechanism of tumor growth, diffusion and implantation within the mesothelial layers are less studied when compared to other types of peritoneal malignancies, which are more frequent and characterized by higher cellularity [6,7,8].

An important role for cancer dissemination into the peritoneal cavity is played by epithelial–mesenchymal transition (EMT) [9,10]. EMT and its reverse, mesenchymal–epithelial transition (MET) are biological processes involved in embryogenesis and organ development (type 1 EMT), tissue regeneration and fibrosis (type 2 EMT) and cancer progression and metastasis (type 3 EMT) [11,12]. More recent studies have demonstrated that tumor cells may show a spectrum of intermediate states, termed transitioning or hybrid states, between the full epithelial and mesenchymal states. Transitioning phenotypes have been shown to be associated with different outcomes in carcinomas and sarcomas [13,14], suggesting that their clinical behavior might be dependent on the tumor type [15]. The dynamic nature of EMT and MET processes was collectively defined as epithelial–mesenchymal plasticity (EMP) [12,16,17]. A consensus statement including the nomenclature and guidelines for EMT research has recently been published [18].

In this review, we discuss the present understanding of the possible involvement of pathways related to EMP in the pathogenesis and clinical behavior of PMP, with a focused perspective on the molecular and biological factors involved in the progression of this tumor that could be targeted to enhance the current therapies.

## 2. Pseudomyxoma Peritonei

### 2.1. Etiopathogenesis

PMP is a clinical condition characterized by intraperitoneal dissemination of mucinous tumors and accumulation throughout the abdominal cavity of mucinous ascites (Figure 1a) originally called “jelly belly”, when first described in 1884 by Werth [19]. In the majority of cases, PMP originates from an appendiceal mucinous neoplasm, but a similar clinical picture has been exceptionally described in some ovarian (mature cystic teratomas), colorectal, pancreatic, urachus and breast neoplasm [20]. In 2–5% of cases the origin remains unknown. In appendiceal tumors, the neoplastic proliferation of goblet cells leads to a constitutive production and accumulation of mucin in the lumen (appendiceal mucocele, Figure 1b), causing the rupture of the appendiceal wall and the spread of neoplastic cells into the peritoneal cavity.

The exfoliated mucinous cells tend to be redistributed into the peritoneum according to abdominal fluid hydrodynamics, which is regulated by gravity and by the diaphragm and bowel movements [2,3]. Mucinous cells tend to implant at reabsorption sites of peritoneal fluids, the so-called lymphatic stomata (also known as the redistribution phenomenon). Therefore, these sites and the dependent recesses of the peritoneal cavity may be invaded by tumor implants and mucin. The progressive accumulation of mucinous ascites is favored by the absence of symptoms in the early phase of the disease. This condition invariably progresses and leads to abdominal distension and compression of visceral organs, causing the majority of symptoms, disability and complications in affected patients [21]. Moreover, progressive inflammation promotes fibrotic reaction of mesothelium, with the development of intestinal obstruction, which is often a fatal complication of untreated or recurrent PMP [22]. The main steps of PMP pathogenesis are summarized in Figure 2.

### 2.2. Epidemiology and Diagnosis

PMP is considered a rare disease with an estimated incidence rate between 1 and 4 people per million per year [23,24]. The diagnosis is frequently made after the age of 40 with a 2–3 × higher incidence in women [23,25]. Clinical presentation is quite variable [26]. Most patients are diagnosed through a laparotomy/laparoscopy performed for suspected appendicitis or peritonitis or other surgical procedures, such as groin hernia repair. In women, PMP is frequently diagnosed following investigations of an ovarian mass or for infertility. However, a significant number of PMP remains frequently asymptomatic for a long time, becoming evident only after radiological examination performed for other causes. In advanced stage, clinical presentation includes increased abdominal girth or ascites. In patients with suspected PMP, laparoscopy is the most sensitive method for diagnosis and staging. Laparoscopy allows for direct visualization of tumor nodules and for large-size biopsy, as the large areas of acellular tissue leads to a high risk of inconclusive diagnosis following radiologically-guided biopsy [27]. Computed tomography scan is considered the gold standard for PMP diagnosis and staging [28]. T1–T2 weighted magnetic resonance imaging scans are more sensitive in distinguishing between mucin and cellular tissue, while positron emission tomography is in general of little value as mucinous implants have a very low metabolic activity due to the paucity of tumoral cells [29]. Carcinoembryonic antigen (CEA), carbohydrate antigen 19.9 and 125 (CA19.9 and CA125) have a relative role in the diagnostic framework, but are more relevant as prognostic markers and for detecting early recurrence [28,30].

### 2.3. Histologic Variants

The prognosis of PMP is highly dependent on the histological grade, which is a strong prognostic factor in patients with stage IV mucinous carcinoma according to the American Joint Committee on Cancer (AJCC) staging [31]. Outside the AJCC staging system, two main classifications of prognostic value have been proposed, and both distinguish low-grade and high-grade PMP according to defined histological features associated with a clear prognostic value [32]. A significantly worse prognosis is linked to the histological evidence of signet-ring cells within the high-grade PMP group. An attempt to resolve the relative confusion and controversy generated by these classifications (two and three-tier) and the other histologic variants described, a consensus was recently finalized by the Peritoneal Surface Oncology Group International (PSOGI) for a shared classification system suitable for diagnosis and treatment [33]. The PSOGI system divided PMP in which epithelial cells are detectable in three groups according to the histological grade (low-grade, high-grade and high-grade with signet-ring cells). The term “acellular mucin” should be considered in those cases in which tumor cells are not detectable (Table 1).

### 2.4. Molecular Profile of PMP

The rarity of the tumor and the low cellularity make molecular profiling of PMP challenging. Over the last few years, several studies have investigated the potential role of PMP variants for diagnostic purposes, to allow differential diagnoses among diverse mucinous cancers. Moreover, genetic profiles might also be useful for prognostic stratification, thus to possibly facilitate the selection of patients for surgery.

*KRAS* mutations have been reported in up to 94% of primary low-grade appendiceal neoplasia [34,35,36] and in 81–93% of low-grade/high-grade PMP [37,38,39], and therefore *NRAS* and *BRAF* mutations were very seldom detected. Peculiar to PMP seems to be mutations in *GNA*S, which are mostly found, albeit at variable rates (25–100%), in low-grade cases [40]. In colorectal cancer (CRC), *GNAS* mutations are found very infrequently (2.5%), and, when present, the tumor shows peculiar clinical and pathological features resembling PMP of appendiceal origin, i.e., mucinous histology, right side origin, peritoneal metastases and association with *KRAS* mutations. The frequent coexistence of *KRAS* and *GNAS* mutations (>70%) in mucinous appendiceal tumors suggests a possible molecular interaction of these two genes in PMP tumorigenesis [39,40]. For prognostic stratification, *KRAS* mutations are significantly correlated with progression-free survival after CRS/HIPEC, while the role of *GNAS* is more controversial [38,41]. *GNAS* mutations seem to be related more to mucin production rather than tumor proliferation.

The functional loss of the DNA mismatch repair (MMR) pathway, which is involved in maintaining genome stability and integrity, is a rare event in PMP. This condition determines the accumulation of mutations in oncogenes and tumor suppressor genes and causes microsatellite instability. It is estimated to be present in 6.3% of PMP cases and is related to a worse prognosis [42].

### 2.5. Mucins in PMP

Mucins are high-molecular-weight O-glycoproteins composed of a long polypeptide chain on which several oligosaccharide chains are linked. They are produced by the secretory epithelium in both secreted and membrane-bound forms. Secreted mucins are arranged as networks of monomers forming homo-oligomeric structures with viscoelastic properties, whereas membrane-bound mucins are monomeric and do not form gels. Under physiological conditions, mucins exert a protective function on mucosal surfaces as well as modulate cell-to-cell and cell-to-matrix interactions, epithelial cell growth and differentiation [43]. The aberrant expression of mucins in terms of quantity or tissue distribution is involved in several pathologic conditions, including cancer [44]. In PMP, deposits of mucin can be acellular, or organized with capillaries and other stromal and immune cells around tumor nodules implanted on the mesothelium. Moreover, single or small clusters of neoplastic cells may circulate in the peritoneum with a mucin coat.

Three extracellular, gel-forming types of mucin are mainly found in PMP secretions: MUC2, MUC5AC and MUC5B [45]. The presence of MUC4, a membrane-bound mucin, was reported in a single case [46]. MUC2, which is predominantly expressed in the gastrointestinal tract, is recognized as a biomarker for PMP of appendiceal origin [47,48,49]. MUC5AC is physiologically found in the respiratory tract, stomach and gynecologic sites (endocervix and endometrium) and is present in many tumor types. The relative abundance of MUC2 compared to MUC5AC may be used for differential diagnosis with mucinous implants of ovarian cancer (OC), in which MUC5AC is, in most cases, the predominant mucin. Thus, MUC2 predominance indicates a likely intestinal origin [47]. Both MUC2 and MUC5AC, which are secreted by goblet cells [50], are transcriptionally regulated by several molecules such as proinflammatory cytokines (interleukin-1beta (IL-1*β*), IL-6 and tumor necrosis factor-alpha (TNF-*α*), pleiotropic cytokines (such as IL-9), bacterial exoproducts, growth factors, retinoids and hormones [51]. Among all these factors, IL-9 was found to be predominantly detected in PMP compared to adenocarcinoma [49]. The third most frequently detected mucin in PMP is MUC5B, which is primarily expressed in the conducting airways, salivary gland and endocervix. MUC5B exists in two different glycosylated variants, a high-charge and a low-charge glycoform [52]. The amount of the low-charge glycoform of MUC5B might be responsible for the increased level of hardness of mucin found in some PMP patients [45,53]. Based on the different proportions of the three mucins and according to their physical and chemical properties, three grades of PMP mucin secretions (soft, semi-hard and hard texture) were characterized [53]. Knowledge of the differential distribution of the three mucins has potentially important implications for the formulation of ad hoc mucolytic combinations that may expand the locoregional therapeutic approaches of PMP.

Mucin overexpression and the contribution of secreted mucins in certain malignancies are in line with the clinical observation of a poor responsiveness of PMP to systemic drugs. Indeed, the mucin/cell ratio is estimated to be higher than 10:1 and the coat of gel-forming mucins seems to act as a protective barrier against chemotherapeutic drug penetration and activity [47].

In vitro studies show that MUC2 overproduction by epithelial cell lines is related to inflammatory or infectious stimulus mediated by the mitogen-activated protein kinase (MAPK) pathway as detailed hereafter [54]. The role of bacteria in PMP is confirmed in the clinical setting, as enteric bacteria have been frequently detected in specimens and free mucin collected during surgery. Bacterial density is significantly correlated with MUC2 expression and is higher in PMP with high-grade histology [55]. The fact that antibiotic treatment improves histopathology and decreases bacterial density suggests a potential role of these drugs against cell proliferation and mucin production [56]. Coherently, proinflammatory factors (EGF, TGF-*α* and TNF-*α*) are able to produce in PMP cell lines a significant increase in expression of MUC2 and MUC5AC through activation of the EGFR/Ras/Raf/extracellular signal-regulated kinase (ERK)-signaling pathway [57]. Treatment with anti-inflammatory drugs has been shown to control MUC2 secretion in a PMP mouse model suggesting a potential role of corticosteroids and COX-2 inhibitors in disease control [58]. Hypoxia was also found to induce MUC2 expression, through upregulation of hypoxia-inducible factor-1alpha (HIF-1*α*) resulting in increased binding of HIF-1*α* to the MUC2 promoter. MUC2 expression was found to be reduced in vivo by HIF-1*α* inhibitors such as BAY 87-2243, and MUC2 reduction significantly prolonged animal survival [59]. Moreover, inhibition of the MAPK pathway with specific targeted drugs gave promising results in the same PMP xenograft model, suggesting potentially effective future strategies for the control of mucin production and tumor growth in PMP [60].

As discussed above, *GNAS* is mutated in a proportion of PMP [40]. *GNAS* encodes Gsα, the G-protein α-subunit that transduces signals from G-protein-coupled receptors (GPCRs) to adenylyl cyclase, and regulates the expression of cyclic adenosine monophosphate (cAMP). The mutation sites of *GNAS* in PMP alter the structure and the enzymatic activity of the GTPase domain of the α-subunit, and determine the continuous binding of GTP to Gsα. The resulting constitutive activation of this subunit stimulates the cAMP-PKA signaling pathway, which is involved in mucin gene expression [61,62]. A CRC cell line stably transfected with a *GNAS* mutant showed elevated cAMP levels and a significant increase in mucin production not paralleled by a change in cell proliferation, a result that accords with the clinical behavior of PMP [34,40]. However, mucin production is also altered in *GNAS* wild-type patients, indicating the involvement of alternative pathways in mucin overexpression.

### 2.6. Current Therapies

Historically, debulking surgery has been the mainstay of PMP treatment and the only available therapy able to guarantee long-term tumor control [63,64]. In the early nineties, an innovative locoregional approach was proposed by Sugarbaker [65], based on a more aggressive cytoreductive surgery (multiple peritonectomies and visceral resections) combined with hyperthermic intraperitoneal chemotherapy. CRS/HIPEC has been shown to significantly improve disease control and overall survival, and it should be considered the standard of care for the majority of PMP patients within centers with experience in peritoneal tumor treatment [5]. CRS/HIPEC achieves currently the best locoregional disease control, with a median overall survival and progression-free survival rate of 16.3 years and 8.2 years, respectively [5]. In patients with recurrent and/or unresectable disease, surgery remains the best therapeutic option to palliate obstructive symptoms such as abdominal discomfort and pain. More recently, an innovative locoregional approach has been developed, the pressurized intraperitoneal aerosol chemotherapy (PIPAC), which has been introduced as palliative treatment for patients with unresectable peritoneal metastases [66]. PIPAC is a minimally invasive surgical technique for intraperitoneal drug delivery able to treat peritoneal cancer nodules of different origin (gastric, colorectal and ovary). It was shown to increase drug penetration without altering the distribution pattern in a post-mortem swine model [67]. Although rarely used in PMP, it represents a promising approach for disease control in cases not eligible for radical surgery.

Systemic chemotherapy should be considered when CRS/HIPEC is not indicated due to comorbidities or unresectable disease, or in the case of recurrent PMP, for which iterative CRS/HIPEC is ineffective. However, limited data are available on the role of chemotherapy in PMP patients, who are generally considered unresponsive to standard 5-fluoropyrimidine-based regimens [68]. Even if the response rate in these cases is expected to be no more than 20%, systemic chemotherapy is the only treatment available to realistically obtain disease control in patients who are not candidates for CRS/HIPEC. The frequent *KRAS* mutations in PMP predict that therapy with epithelial growth factor receptor (EGFR)-targeted agents, as demonstrated in CRC, is likely to be ineffective [69]. Likewise, BRAF-targeted therapy seems unlikely to be effective, as *BRAF* is rarely mutated in PMP.

Although the recent introduction of immunotherapy has shown promising results in CRC patients [70], especially in patients with defective DNA MMR, this treatment has not yet been tested in PMP patients for the rarity of this functional loss. An ongoing phase II study of nivolumab and ipilimumab is recruiting patients with metastatic mucinous colorectal and appendiceal tumors with proficient DNA MMR (ClinicalTrials.gov Identifier: NCT03693846). The potential effect of innovative locoregional treatments, such as PIPAC, in eliciting an immune response by increasing the release of tumor-associated antigens is presently under active investigation.

In patients with extensive or recurrent disease, one strategic target for therapy is to promote breakdown of mucin to control symptoms and improve the quality of life in PMP patients. Among different mucolytic agents, bromelain and N-acetylcysteine have been tested with encouraging results. The activity of these two drugs was analyzed on mucinous ascites from PMP patients and on mucin-producing gastrointestinal cancer cell lines [71,72]. Their combination showed a synergistic effect in the mucolytic activity in a rat preclinical model intraperitoneally implanted with mucin. Moreover, they showed antiproliferative effects in a nude mouse model xenografted with mucin-producing cell lines [71,72]. These results were confirmed in a phase I trial in humans, where a significant mucolytic activity was detected after percutaneous injection, with a good radiological response on mucin accumulation and irrelevant toxicity [73]. A multicenter trial investigating the effectiveness of bromelain and N-acetylcysteine (BromAc) in patients with mucinous tumors including PMP, administered directly into the tumor or in the peritoneal cavity through a percutaneous drain, is presently ongoing (ClinicalTrials.gov Identifier: NCT03976973). Cysteamine, another mucolytic substance, was analyzed in combination with bromelain and showed promising results in an in vitro system simulating a peritoneal wash [53].

## 3. Epithelial–Mesenchymal Plasticity and Peritoneal Dissemination

### 3.1. Current Evidence of EMP and Hybrid States in Peritoneal Metastases

Type 3 EMT is associated with the tumor-initiating and metastatic potential of neoplastic cells [11]. EMT confers an increased resistance to chemo- and immunotherapy to neoplastic cells, a property interwoven with stemness, as EMT contributes to the generation and/or maintenance of cancer stem cells (CSCs), a small subset of cells with self-renewal capacities that are also responsible for tumor cell drug resistance [74,75].

The activation of EMT induces the loss of apical-basal polarity and cell–cell epithelial interactions, followed by the rearrangement of the actin cytoskeleton, and, when fully executed, leads to a mobile, matrix-interacting mesenchymal cell. Mesenchymal cells are able to express several matrix metalloproteases (MMPs), which allow them to degrade and invade the basement membrane and extracellular matrix (ECM). All these changes, relevant to the transcoelomic dissemination of the tumor, are accompanied by molecular hallmarks, including the sequential loss of epithelial components involved in epithelial junctions, the main being E-cadherin, in addition to β-catenin and Zonula Occludens 1 (ZO-1). This is paralleled by the gradual increment of mesenchymal factors, such as vimentin, N-cadherin, α-smooth muscle actin (α-SMA), β1 and β2 integrins and MMPs. This process is coordinated by a complex network of factors and pathways, including transcription factors (ZEB1/2, SNAIL1/2 and TWIST1/2) and other complex regulatory levels, activated by a plethora of extracellular signals [75,76]. The resulting transition is a highly flexible phenomenon, summarized as EMP, as it is influenced by the cellular and environmental context. It has become evident that each transition is not a simple shift from a full epithelial to a full mesenchymal state and vice versa. Instead, cells can proceed into a spectrum of intermediate/hybrid states, which may be stable, eventually representing the last step of this process, or metastable, thus cycling through rounds of EMT and MET in a context-dependent manner. Intermediate states possess various combinations of epithelial and mesenchymal components, and tumors, temporally and spatially, may exhibit different transitioning EMT states, which may govern the clinical outcome [12,15,17].

OC represents a well-known model of peritoneal dissemination in which hybrid EMT states were first recognized [10,77]. During progression, EMT-driven delamination of OC cells from the primary tumor occurs, and cellular clusters and/or single cells display transcoelomic dissemination. EMT is triggered and/or maintained by different stimuli in ascites, mainly through TGF-β [78], which confers anoikis-resistance to single cells. Cells can further avoid the surrounding stresses by forming spheroids. Spheroids were the first tumoral entities to exhibit a hybrid phenotype in clinical samples of malignant ascites and in OC cell lines in low-attachment culture conditions, as reviewed by Davidson and colleagues [77]. In vitro experiments showed that spheroids could clear the mesothelial cell (MC) layer, disaggregate, attach to and invade ECM components [79,80], thus generating the site for a secondary metastatic tumor [81]. Molecular analyses of clinical samples and OC cell lines generated conflicting data concerning the presence and prognostic significance of EMT biomarkers in primary tumors, ascites and peritoneal implants [77]. Nevertheless, it was consistently shown that effusional spheroids may maintain different levels of E-cadherin expression in a mesenchymal background [77]. Spheroids were also shown to include a subset of CSC able to reproduce the heterogeneity of the tumor of origin and to provide a chemoresistant subset that constantly repopulates the abdominal cavity. Using in vitro spheroid formation assays and EMT induction through TGF-β treatment, spheroids were shown to be enriched in stemness markers, such as pluripotent stem cell transcription factors NANOG, OCT4 and SOX2, and stem cell surface markers, such as CD117 and CD133 [82]. Overall, a spectrum of EMT states was found in spheroids and OC cell lines, which was associated not only with different degrees of chemoresistance and with a stem-like phenotype, but also with overall survival and disease relapse [81,83,84], indicating that modulation of epithelial plasticity is implicated in the biological and clinical behavior of OC.

Peritoneal involvement is found in about 5% of CRC patients as a unique metastatic site, and, alone or in the presence of other metastatic sites (liver and/or lung), represents an unfavorable prognostic factor [85,86]. Abdominal spread can occur from EMT-induced primary tumor cell detachment, and after surgical seeding when the resection margins are close to the primary tumor and/or after lymphatic/blood vessels transection. Several studies have focused on EMT and metastatic spread in CRC [9,87] and will not be discussed in this review. The degree of invasion of the peritoneal elastic lamina, a network of elastic fibers located in the submesothelial region, was studied in 564 CRC cases and was associated with peritoneal dissemination and distant metastases [88]. The subserosal invasive front was characterized by a high number of tumor buddings. Indeed, an interesting EMP-related feature of CRC, with unfavorable prognostic significance, is tumor budding, reported in 20–40% of tumors, which has more recently been linked to epithelial–mesenchymal plasticity. Tumor budding is defined as a single tumor cell, more rarely, or a cluster of up to five neoplastic cells observed closely at the invasive front of the tumor. Collective cell migration is also found in other tumors, and thought to represent the histological image of a hybrid state, connecting the epithelial feature of cell adhesion to the migratory capacity of mesenchymal cells [89,90]. Major inducers of EMT in tumor buds are secreted by tumor-associated stromal components; among these inducers are the hepatocyte growth factor (HGF), EGF and TGF-β, which activate intracellular EMT networks involving SMADs, ERK, MAPK and PI3K/AKT signaling pathways. E-cadherin usually sequesters β-catenin at sites of cell–cell contacts; decreased E-cadherin expression during EMT leads to nuclear translocation of β-catenin and activation of WNT/β-catenin signaling, shown to be a relevant pathway in tumor budding in CRC patients [89].

Immunohistochemistry (IHC) in resected CRC showed, in tumor buds, a decrease in the expression of membrane-associated E-cadherin and β-catenin complexes, with a strong reactivity of β-catenin in the cell nuclei. Moreover, staining for pan-Cytokeratin, more dynamic epithelial components, and a concomitant increase in MMPs and other enzymes involved in ECM degradation were observed. Tumor buds, like spheroids, are also characterized by stem cell markers, such as CD133 and aldehyde dehydrogenase 1 [89,90,91], providing further evidence of the role of epithelial plasticity in stemness.

### 3.2. Mesothelial Cells and Tumor Progression

Serosal body cavities such as the peritoneum are lined with MC, which provide a frictionless surface for the free movement of abdominal organs and a first protective barrier for inflammation, infections and tumor invasion. MCs are extremely plastic cells, as they may undergo EMT and convert to myofibroblasts, downregulating cytokeratins and E-cadherin, and upregulating α-SMA and Vimentin. This phenomenon is implicated in wound healing and organ fibrosis and it is related to type 2 EMT [92]. MC and their transdifferentiated counterparts are also involved in tumor progression, as they were shown to contribute to OC metastasis by secreting Fibronectin [93], and to support the progression of primary effusion lymphoma (PEL) [94], a non-Hodgkin’s lymphoma that primarily grows as recurrent effusions in large body cavities [95]. The crosstalk between MC and PEL cells, analyzed in coculture systems, showed that PEL cells induce type 2 EMT in MC by secreting TGF-β; MC and myofibroblasts, in turn, were found to confer a growth advantage and increased survival to PEL cells [94], thus favoring lymphoma progression. Moreover, a progressive thickening of serosal membranes was observed in a xenograft mouse model of peritoneal PEL, confirming that fibrosis occurred during intracavitary PEL development [94]. Interestingly, the specific targeting of the intracavitary microenvironment was demonstrated to exert a significant antineoplastic activity in a preclinical PEL/SCID mouse model [96], suggesting that the modulation of mesothelium may offer new therapeutic approaches for primary intracavitary tumors and peritoneal metastases. The role of mesothelium in PMP pathogenesis needs to be fully investigated, although it is feasible that, in late stages, PMP tumor cells, surrounded by a large amount of mucin, are largely isolated from the possible positive or negative signals deriving from the abdominal microenvironment.

### 3.3. Role of Mucins in EMP and Stemness

Some evidence shows a mutual interaction between mucin expression, EMP state and tumor progression. An oncosuppressor activity has been assigned to MUC2, as loss of *MUC2* expression was associated with poor outcome in CRC patients [97,98]. MUC2 acts as caretaker of the epithelial state in colon cancer HT-29 cells, thus inhibiting EMT and metastasis. Indeed, MUC2 suppression, while not affecting cell proliferation, increases cell migration in vitro and promotes liver metastasis in a NOD/SCID mouse model. *MUC2* silencing was shown to lead to downregulation of E-cadherin expression. The aggressive behavior of *MUC2*-silenced cells was shown to be induced by an IL-6-mediated EMT occurring through STAT3 activation [98]. This is reinforced by the evidence that MUC2 and E-cadherin were found to be scarcely expressed in tumor buddings in CRC, particularly at the leading edge of the invasion. On the other hand, tumor budding cells were shown to be enriched in Cytokeratin 7 and maintained Cytokeratin 20. This was initially considered as a modification of EMT into an epithelial–epithelial transition, in which E-cadherin is replaced with other epithelial proteins, the intermediate filaments, representing more dynamic elements of the cytoskeleton [99]. What was initially considered an aberrant expression could now be represented as a transitioning state. These data highlight that further studies are needed to characterize all steps involved in epithelial plasticity, including not only the coexpression of epithelial and mesenchymal proteins, but also the switch of an epithelial, static state to a similar but more dynamic one, with de novo induction or upregulation of *KRT* expression.

Among the oncogenic mucins, MUC5AC was shown to be overexpressed in pancreatic cancer cell lines by GLI1- and GLI2-mediated transcriptional activation and to increase migration and invasion of pancreatic neoplastic cells by localizing in the intercellular junctions and interfering with E-cadherin membrane localization [100]. IHC studies conducted on clinical pancreatic tumor specimens confirmed the correlated expression of GLI1 and MUC5AC [100]. Accordingly, overexpression of MUC5AC, along with MUC1, in colon cancer cell lines increased their proliferation, migration and invasive potential [101,102] and was associated with a decreased expression of E-cadherin [101]. Another mucin found in PMP, MUC5B, was shown to be involved in the increased proliferation and invasiveness of breast cancer MCF7 cells in vitro, and promoted tumor growth and metastases in a xenograft model [103].

Some mucins are involved in the induction and/or maintenance of CSC. Indeed, MUC4-downregulation in pancreatic CSC was shown to reverse chemoresistance and prevent tumor relapse [104,105]. MUC4 overexpression is responsible for the enrichment in the CSC subset in OC cell lines [106]. MUC5AC was shown to interact with CD44 and to induce chemoresistance in CRC through β-catenin/p53/p21 signaling pathway. This interaction was demonstrated to contribute to the maintenance of stemness and spheroid-forming ability of the side population in CRC cell lines [102].

## 4. Clinical and Molecular Evidence of Epithelial Plasticity in PMP

### 4.1. EMP Markers and Signatures in PMP Cells

Epithelial plasticity is likely to be implicated in the invasive capability of PMP cells, and biomarkers linked to a transition towards a mesenchymal state were shown to have a negative prognostic significance. In fact, early studies identified a pattern of expression indicative of a more mesenchymal state in diffused PMP, characterized by reduced E-cadherin and increased N-cadherin reactivity [49]. This switch in *CDH1* expression was also evidenced in advanced stage CRC [49,107], and was linked to tumor dissemination in the abdominal cavity in PMP.

Additional IHC studies compared the expression profile of epithelial and mesenchymal markers on isolated, scattered vs. cohesive groups of tumor cells found in mucinous ascites and peritoneal implants from the two main prognostic groups, DPAM and PMCA [20,108]. Single cells were shown to be immunonegative for E-cadherin and β-catenin, with strong Vimentin reactivity in almost all DPAM and in all PMCA samples, whereas cell groups showed the opposite pattern of staining (Figure 3). Moreover, single cells were more likely to have decreased expression or to be immunonegative for Cytokeratin 20 and/or Cytokeratin 7 compared to cell clusters, which showed reproducible reactivity for Cytokeratin 20. In light of the more recent concept of EMP, these data might suggest that single PMP cells likely display a more advanced EMT state compared to cell clusters, and thus may be responsible for the more aggressive mucinous invasion of the abdominal cavity. Indeed they were found more frequently in PMCA than in DPAM [20,108], and a higher number of single cells was correlated to a faster progression [108]. Moreover, disseminated single cells frequently show signet-ring differentiation, with a clear negative impact on prognosis [20].

How EMT state is regulated at a molecular level in PMP is still under study. Much evidence shows that, within low-grade PMP, clinical outcomes exhibit a significant variability [109,110]. Global gene expression analyses carried out on low-grade PMP undergoing CRS/HIPEC distinguished two molecular subtypes with diverse prognosis. Interestingly, an EMT signature was found among the highly enriched gene sets in the poor prognosis subgroup, suggesting that pathways involved in epithelial plasticity might contribute to the aggressiveness of a subset of low-grade PMP [111].

Roberts and colleagues [8] showed upregulation of *DSC3* expression in PMP compared to normal mucosal epithelium. Desmocollin 3, encoded by *DSC3*, is a member of the *CDH1* gene superfamily coding calcium-dependent cell adhesion molecules and a component of desmosomes, intercellular junctions that are usually present on the lateral side of polarized epithelial cells and linked to the intracellular intermediate filaments [112,113]. This transmembrane adhesion protein is involved in tumor pathogenesis due to its ability to inhibit cell motility, thus acting mainly as a tumor suppressor. Indeed, it is frequently downregulated or poorly expressed in breast, lung and colon carcinoma [114,115,116]. On the other hand, in certain cancers, overexpression of desmosome components is linked to tumor progression [113,117]. The role of this protein in PMP pathogenesis needs to be further investigated.

### 4.2. Dysregulation of EMT-Related Pathways in PMP

The TGF-β/SMAD, WNT and PI3K/AKT signaling pathways play a relevant role in determining the plasticity state in several tumors. TGF-β exerts its activity through binding to cell surface TGF-β type I and II receptors (TβRI and TβRII), which form a heterotetrameric complex in which the active TβRII kinase domain phosphorylates TβRI, thus activating internal cascades through SMAD and non-SMAD signaling pathways [118]. In the canonical SMAD pathway, TβRI recruits and phosphorylates specific receptor-regulated SMADs (R-SMADs), such as SMAD2 and SMAD3, that form heterodimeric complexes with common SMADs (Co-SMAD), such as SMAD4 [119].

The relevance of the TGF-β/SMAD pathway in PMP pathogenesis was identified following the detection of mutations in its components in a small group of PMP patients. *TGFBRI*, *TGFBRII*, *SMAD2*, *SMAD3* and *SMAD4* were the most frequently mutated members of this pathway in PMP [37,120]. Mutational inactivation of the TGF-β receptors and SMADs was also detected in advanced adenomas and affects half of all CRC [121,122]. Among SMAD proteins, SMAD4 plays an important role in epithelial–mesenchymal plasticity. In CRC, SMAD4 inactivation was found to lead to aberrant activation of STAT3, which is a well-known signaling pathway involved in EMT activation through ZEB1 [123]. Moreover, SMAD4 loss was associated with worse clinical outcome and increased chemotherapy resistance in CRC patients [124]. The role of these mutations as prognostic markers in patients affected by PMP needs to be further investigated.

The role of other dysregulated pathways in PMP pathogenesis was suggested by the evidence of *CTNNB1* mutations affecting the WNT pathway [37,38], and *AKT1* and *PI3KCA* mutations of the PI3K/AKT pathway [38,125,126].

### 4.3. Mucins and EMP: Is There Any Link in PMP?

Although the three mainly secreted mucins by PMP were found to be associated with biological properties and molecular pathways linked to epithelial plasticity in other tumors, it remains to be demonstrated whether they play a role in EMP of PMP cells. A possible link is suggested by IL-9, which promotes *MUC2* expression. Interleukin-9 and its receptor (IL-9Rα) were found highly expressed in PMP compared to colon adenocarcinoma [49]. This interleukin might thus exert a pathogenic role in establishing an autocrine loop leading to mucin overexpression, and, possibly, transformation of goblet cells. Interestingly, in pancreatic cancer cell lines, IL-9 was shown to stimulate proliferation, migration and invasion of tumor cells by decreasing a well-known EMT inhibitor, miR-200a [127]. Similarly, in non-small cell lung cancer cells, increased IL-9 was found to counteract the miR-208b-5p-mediated suppression of EMT by inactivating the STAT3 signaling pathway. In parallel, miR-208b-5p-overespressing cells exhibited a typical epithelial state, with increased E-cadherin expression and suppression of N-cadherin and Vimentin [128]. These data suggest that IL-9 might also act as an EMT inducer in PMP, possibly by counteracting EMT suppressors such as the miR-200 family and enhancing the expression of EMT-inducing mucins. However, direct evidence of these possible interactions is not yet available.

Furthermore, while MUC2 was shown to be involved in the control and maintenance of the epithelial state in CRC, PMP cells are not constantly kept in an epithelial state in the presence of MUC2. Indeed, PMP samples were shown to be mainly immunoreactive for MUC2 and Cytokeratin 20, with variable degrees of expression of E-cadherin and Cytokeratin 7 [49,129]. Moreover, isolated PMP cells in mucin pools are more likely to be E-cadherin^-^/Vimentin^+^, and immunopositive for MUC2 and Cytokeratin 20, therefore displaying an expression pattern of cells transitioning towards a more mesenchymal state [108]. These findings might suggest that more complex and context-dependent mechanisms of EMP modulation act in this pathological condition. Molecular data dissecting the link between EMP and MUC2 are currently lacking.

### 4.4. EMP and Heterotopic Ossification in Epithelial Tumors

Tumors of epithelial origin may, very rarely, show the presence of sites of ossification near specific cells that express typical bone-specific markers, such as vitamin D receptor, RUNX2 and RANKL. These cells possess the ability to generate areas of ossification, thus fully mimicking the phenotype and function of osteoblasts. These osteoblast-like cells have been detected, albeit rarely, in different types of epithelial tumors, such as breast, hepatocellular and gastrointestinal carcinomas [130]. It has been shown that heterotopic ossification is generally associated with the expression of mesenchymal markers, such as Vimentin and β-catenin, implying that EMT is involved in this pathologic feature [130].

Different pathogenetic mechanisms have been hypothesized. Tumor cells, in an autocrine TGF-β/bone morphogenetic protein (BMP)-mediated loop, may undergo an EMT and, after acquisition of a mesenchymal state, may differentiate into osteoblast-like cells. Concomitantly or alternatively, neoplastic cells, through paracrine stimulation, may induce osteogenic metaplasia in stromal cells of the tumor microenvironment. These hypotheses were formulated on the basis of IHC analyses that found evidence of bone-specific proteins in tumor and stromal cells in breast tumors and gastrointestinal cancers [130,131].

Interestingly, osteoblast-like cells were found in other tumors showing high frequency of bone metastases. It was hypothesized that these cells, adapted to a bone-like microenvironment and thus exhibiting a peculiar osteotropism, might have a prognostic significance for their ability to promote distant bone metastases [132].

Two cases of perforated low-grade appendiceal mucinous carcinomas and peritoneal diffusion with multifocal calcified portions have been reported [133,134]. A search for osteoblastic markers, specifically BMP9, Osteocalcin (or Bone γ-carboxyglutamic acid-containing protein, BGLAP) and Osteopontin (OPN), has been performed in one case of PMP [134]. BMP9 is one of the most potent osteogenic factors not only because it is a strong activator of mediators of osteogenic signaling, but also because it is involved in a network of interactions with several signaling pathways [135]. Osteocalcin is secreted by osteoblasts and acts mainly in an endocrine manner controlling several physiologic processes, having a minor role in determining bone density and mineralization [136]. On the other hand, OPN is a matricellular glycoprotein highly expressed in osteoblasts and osteoclasts and it is critically involved in biomineralization [137]. IHC analyses in one PMP case showed that BMP9 and osteocalcin were found in tumor cells, osteoblasts and stromal cells, whereas OPN was detected only in the cytoplasm of tumor cells [134]. The presence of the three bone markers in tumor cells might thus reflect the activation of tumor plasticity pathways promoting the transition towards an osteoblast-like phenotype. Moreover, as BMP9 was also detected in stromal cells, tumor cells might activate an interactive crosstalk with the tumor microenvironment. Therefore, the dynamic interaction between tumor and stroma might contribute to inducing osteoblastic differentiation of stromal mesenchymal cells. Figure 4 shows the possible pathogenic mechanisms involved in this phenomenon.

Signaling pathways involved in PMP ossification are unsurprisingly connected with EMP, as bone morphogenetic proteins are multifunctional cytokines belonging to the TGF-β family. BMPs play an important regulatory role in early mesenchymal stem cell differentiation, in osteoblastic differentiation and in bone induction, maintenance and repair [138,139]. As TGF-β, the BMP signaling pathway engages two cell surface BMP receptors, which are serine-threonine kinase receptors, that form heterodimeric complexes [140]. They transmit the signal through SMAD-dependent or -independent mechanisms, the latter involving MAPK signaling [141].

In the canonical SMAD-dependent signaling pathway, the activated receptor complex phosphorylates the carboxy-terminus of the R-SMADs, which interact with different downstream proteins, including RUNX2. In fact, upon activation of BMP signaling cascade, RUNX2 and SMAD physically interact to cooperatively regulate the transcription of target genes. RUNX2 is an essential transcription factor for osteoblast differentiation and induces the expansion of osteoblast progenitors by regulating the expression of *FGFR2* and *FGFR3* genes [142,143]. Therefore, the coordinated activity of BMP-activated SMADS and RUNX2, critical for bone development and regeneration, is also involved in extraskeletal ossification.

In addition to the canonical SMAD-mediated signaling pathways, BMP can activate several MAPKs, including ERK and p38 kinases. It has been shown that several positive, negative or synergistic effects are induced when the TGF-β/BMP pathway interacts with proteins of the MAPK, WNT, Hedgehog (Hg), NOTCH and AKT/mTOR pathways, to regulate BMP-induced signaling. While RUNX2 is a key integrator between the pathways, Hg acts as modulator [139]. The mechanisms implicated in dysregulation and abnormal expression of BMPs by tumor cells are not fully understood. However, proteins of the Hg pathway, likely engaged during the process of EMT, were found to be correlated with BMP signaling by increasing the transcription of the *GLI2* gene, which encodes a zinc finger transcription factor strongly activated by the TGF-β pathway. GLI2 is in turn a potent cis-activator of *BMP* gene expression, suggesting that BMPs and GLI2 can establish an autostimulatory loop [144]. Interestingly, it was shown that activation of Hedgehog signaling caused by GNAS inactivation leads to heterotopic ossification [145], suggesting a possible role for other *GNAS* mutations in this phenomenon found in PMP.

## 5. Future Perspectives and Conclusions

PMP is a rare condition with a unique clinical course and fatal prognosis in a significant percentage of patients. The tendency of this disorder to remain confined to the peritoneal cavity and the relative unresponsiveness to systemic and/or targeted therapies has led to the adoption of locoregional treatments such as CRS/HIPEC, which currently allows an optimal disease control and cure in the majority of affected patients. However, some patients are not eligible for surgery due to the presence of extensive or recurrent disease, and they experience a very poor quality of life. Therefore, a better understanding of the molecular basis of PMP progression is urgently needed to increase the chances of treatment and cure in these patients, to improve the results of the CRS/HIPEC in potentially curable patients, and to potentially expand treatment to a wider range.

Mucin heavily affects systemic drug activity and is a determining factor in PMP morbidity. Inhibition of mucin production, through EMP modulation or inhibition of the cAMP-PKA pathway, and the decrease of mucin hardness should be considered for two main reasons. First, the development of preoperative mucolytic protocols in patients selected for potentially curative CRS/HIPEC might make surgery easier and improve surgical outcome by reducing the risk of post-operative complications. Second, mucolytic treatment might also be used for the control of symptoms in patients with unresectable PMP.

A considerable body of evidence shows that EMP is implicated in PMP aggressiveness. Isolated PMP cells have been found to express a transitioning, more mesenchymal profile compared to that expressed by cell clusters, which are characterized by a more epithelial state. The detection of these isolated cells with a more mesenchymal hybrid state in mucin has been associated with a poorer prognosis. Furthermore, an EMT signature is significantly associated with a subgroup of low-grade PMP patients with a more aggressive clinical course. These data clearly highlight the high plasticity of PMP cells, and suggest that factors and expression profiles linked to EMP might have a valuable prognostic significance. In-depth analysis of the prognostic impact of EMP profiles in PMP might permit, in the next future, a better stratification of low-grade PMP patients, by possibly identifying subjects who could benefit in terms of survival from locoregional treatment. EMP is also linked to extraskeletal ossification, a rare pathological feature found in many cancers, including PMP, and very likely to the overexpression of mucins, although this last association has not been directly investigated yet. Dissecting the molecular networks of these interactions might allow for the identification of therapeutic strategies able to interfere not only with PMP aggressiveness, but also with mucin production.

## Figures and Tables

**Figure 1 ijms-21-09120-f001:**
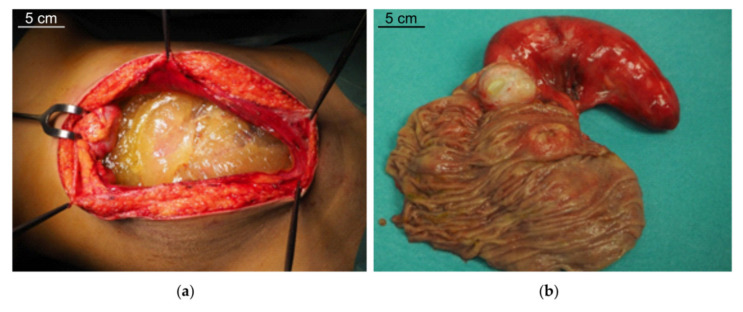
(**a**) Intraoperative view of abdominal mucin accumulation in pseudomyxoma peritonei (PMP) and (**b**) appendiceal mucinous neoplasia with mucocele.

**Figure 2 ijms-21-09120-f002:**
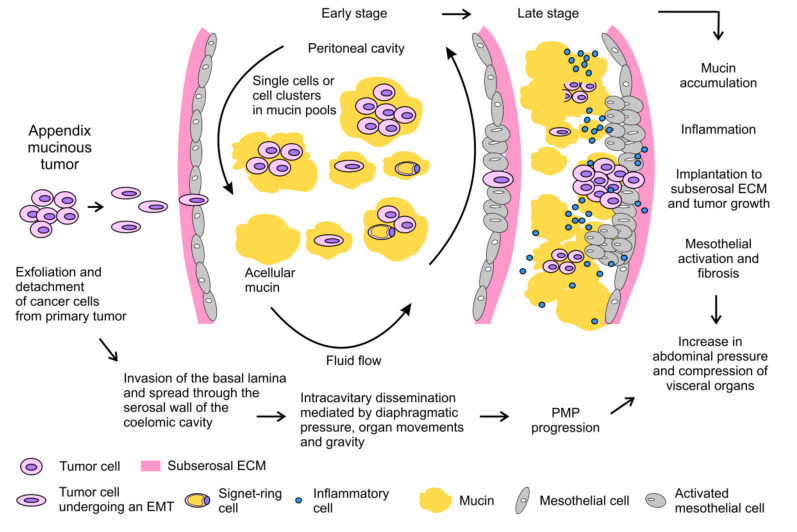
Transcoelomic spread of tumor cells, PMP development and progression. Tumor cells induced to undergo an epithelial–mesenchymal transition (EMT) detach from the primary tumor and invade the peritoneal cavity through the serosal wall. The malignant seeding of mucin-producing single cells and cell clusters into the abdominal cavity is regulated by peritoneal fluid dynamics. Ascitic fluid flow is influenced by the movements of the diaphragm and the abdominal muscles, the peristaltic movements of the intestinal tract and by gravity. The presence of signet-ring cells denotes an aggressive, high-grade PMP. Peritoneal implants are generated by the attachment of tumor cells mainly at sites of fluid reabsorption. The tumor cells invade the subserosal extracellular matrix (ECM) and proliferate. Tumor progression is accompanied by massive mucin accumulation, inflammation and fibrosis of the mesothelial lining.

**Figure 3 ijms-21-09120-f003:**
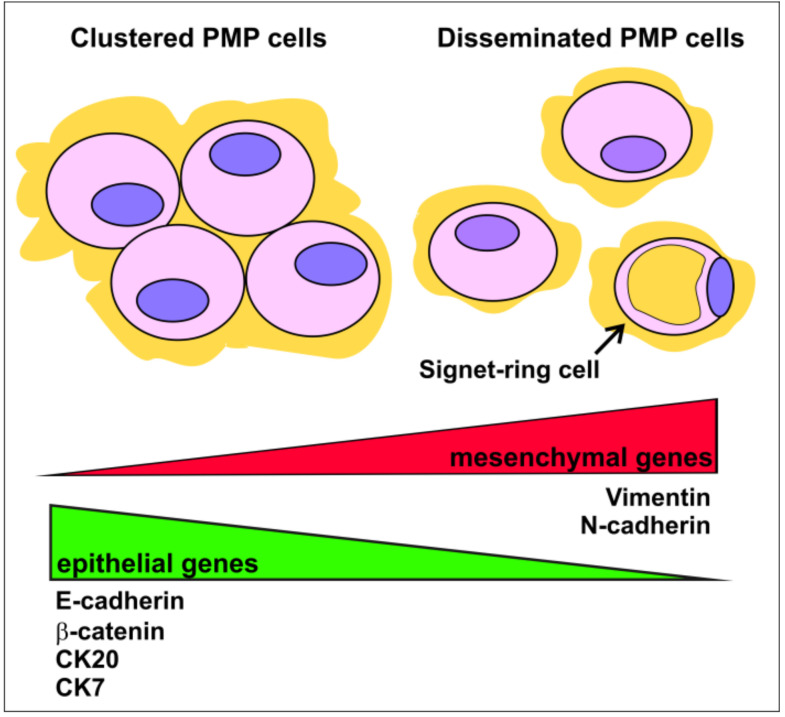
EMT profile in single and clustered PMP cells. Scattered cells in mucin pools are characterized by a more mesenchymal state (red triangle), whereas cell clusters show a more epithelial profile (green triangle). Single cells are frequently associated with a more aggressive clinical behavior, and more frequently show signet-ring morphology, indicated by the arrow. CK: Cytokeratin.

**Figure 4 ijms-21-09120-f004:**
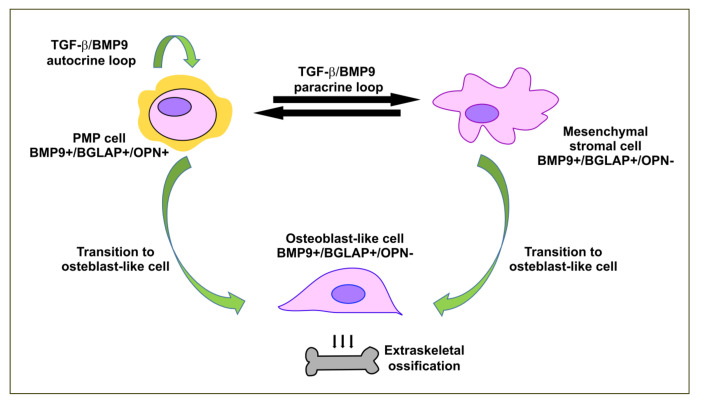
Ectopic ossification in PMP. The tumor microenvironment is composed of cellular and acellular elements, whose interaction may create a dynamic landscape in which EMP takes place. Cytokines and chemokines initially expressed by tumor cells and then induced in the attracted stromal and immune cells may promote cell plasticity. In the context of PMP, TGF-β/BMP9 secreted by PMP cells and expressed by stromal cells may induce EMT in an autocrine and/or paracrine loop. Tumor cells, after having transitioned into a more mesenchymal state, may differentiate towards an osteoblast-like phenotype. In a similar manner, mesenchymal stromal cells, influenced by the same stimuli, may undergo a transition to osteoblast-like cells. These cells were shown to express osteoblastic markers and are very likely responsible for pathologic formation of extraskeletal bone. TGF-β/BMP9: Transforming growth factor-beta/Bone morphogenetic protein 9; BGLAP: Bone γ-carboxyglutamic acid-containing protein or Osteocalcin; OPN: Osteopontin.

**Table 1 ijms-21-09120-t001:** Classification of Pseudomyxoma Peritonei (PMP).

PMP Grading	Current Terminology *	Histologic Features
	Acellular mucin	Mucin with no evidence of epithelial cells
Grade 1	Low-grade mucinous carcinoma peritonei/Disseminated peritoneal adenomucinosis (DPAM)	Pseudostratified or flat strips of epithelium with mild nuclear atypia, pattern of pushing invasion across a broad front and overall maintenance of cellular polarity
Grade 2	High-grade mucinous carcinoma peritonei/Peritoneal mucinous carcinomatosis (PMCA)	Vesicular nuclei with prominent nucleoli, cellular stratification, cribriform or micropapillary architecture, elevated mitotic activity, high cellularity (at least 20% epithelial cells within mucin pools), irregular infiltrative glands or single cell with desmoplasia
Grade 3	High-grade mucinous carcinoma peritonei with signet-ring cells/Peritoneal mucinous carcinomatosis with signet-ring cells (PMCA-S)	High grade histologic features, as reported above, with more than focal areas with signet-ring cell morphology (>10% of cells)

* modified from Carr et al. [33].

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
