# Peer review of "Role of Epithelial–Mesenchymal Plasticity in Pseudomyxoma Peritonei: Implications for Locoregional Treatments"

_ijms, 2020, doi:10.3390/ijms21239120_

Round 1

Reviewer 1 Report

In the review manuscript "Role of epithelial-mesenchymal plasticity in Pseudomyxoma Peritonei: implications for loco-regional treatments”, by Maria Luisa Calabrò et al., the authors combine both clinical and molecular information for a comprehensive overview of PMP pathophysiology and molecular profiling that could improve future PMP diagnosis and treatment, presenting some molecular/mechanistic evidence from other (related) cancer types (e.g. ovarian and colorectal) due to the lack of evidence from PMP. In the authors perspective, the correlation between EMT-associated mechanisms (including extraskeletal ossification), mucins production and PMP aggressiveness should be further elucidated in order to improve current diagnostic and loco-regional treatments.

This review presents a defined focus, in a topic with relevance for cancer research and therapy that lacks substantial revision in the field. The manuscript is well-written, with an eligible English, and generally well-structured and supported by relevant bibliography.

However, in some circumstances, the scientific data described cannot solely be justified by the references used (e.g. reviews) and additional/more relevant bibliography in the field should be chosen to support the mechanistic detail of cell signaling pathways (e.g TGF- TGF-b/SMAD signaling).

In the mechanistic and molecular data, authors should be more consistent and provide whenever possible, more scientific detail on the original studies, as in some circumstances is not possible to discern if the information refers to patients data or experimental models (e.g. cell lines used for in vitro studies or xenograft models, species, cancer type). To improve the scientific significance and quality of the manuscript, it is important to clearly sate if specific mechanisms were already proven to be relevant for PMP progression in patients or are still under scrutiny.

Considering the structure of the manuscript, in order to improve the clarity of presentation and interest for the reader, some revision should also be undertaken. Specifically, in some sections, there is redundant contents presented that could be combined and integrated in unified sections (e.g sections 3.1 and 3.2; sections 4.4 and 4.5 focus in related subjects) and for sake of clarity, information regarding the implication of the same molecule (e.g. Mucins) in PMP should be placed together or in subjacent sections (2.7 Mucins as therapeutic targets and and 3.3 Role of mucins in EMT and stemness).

Overall, I consider that this manuscript provides a comprehensive and significant contribution for understanding the molecular mechanisms implicated in PMP progression that could be further elucidated for improving current diagnosis and treatment, and therefore I consider that it merits publication after minor revisions.

A list of specific concerns and (minor) revisions is provided bellow.

General concerns:

Be more consistent. In some parts of the text is unclear if the data provided or conclusions withdrawn are based on experimental models or patients data (e.g “(…) as they were shown to contribute to Ovarian Cancer (OC) metastasis by secreting fibronectin [75], and to support the progression of Primary Effusion Lymphoma (PEL), (…)” as opposed to other situations were a clear and complete description is provided (“Another mucin found in PMP, MUC5B, was shown to be involved in the increased proliferation and invasiveness of breast cancer MCF7 cells in vitro, and promoted tumor growth and metastases in a xenograft model”).

If possible, be more concise in introductory sections (e.g EMT 3.1) as are supported by review articles as bibliography and part of the information is repeat in subsequent (more subject-oriented) sections (e.g. 3.2.1).

Specific minor revisions:

65: multi-varigated

75: mesenchymal-epithelial transition, already defined before (62)

122: types of mucin (MUC), define first time term is used (line 110)

165: AJCC, no definition in text

173 according the histological grade, to the

176: Table 1. Classification of PMP (modified from Carr [44]), this reference appears twice (also in the footer *modified from Carr et al. [44]).

Define terms/abbreviations (PMP, DPAM, PMCA, PMCA-S) in the footer.

199: defective mismatch repair, dMMR; DNA mismatch repair (MMR) is more commonly used

222: Although the recent introduction of immunotherapy has shown promising results in CRC patients, lacks reference

225: 2.7 Mucins as therapeutic targets, for a better integration of concepts, and to avoid repetition of concepts, consider combining or placing in closer proximity to the 2.2 Mucins in PMP

249: of the MAPK pathway with specific targeted drugs (i.e. MEK inhibitors); unnecessary

255: PMP alter the structure and the enzymatic activity of the GTPase of the a-subunit, unclear, and the enzymatic activity of the GTPase a-subunit (?)

257: In vitro studies; lack detail for the studies referred

265: The two drugs showed in vitro and in vivo in a xenograft model a synergistic effect; lacks scientific detail (cells, animal model)

269: the role of a combination drug therapy with; should reformulate 

274: 3.1 Type 3 EMT and hybrid states: a short introduction, for a better integration of concepts, and to avoid repetition, consider using this introductory info as the first paragraph of section 3.2 Role of EMP in peritoneal dissemination

289: network of factors and pathways, including a relevant group of transcription factors (Zeb1/2, Snail1/2 and Twist1/2), in addition to post-transcriptional, post-translational and epigenetic regulatory; be more concise; post-transcriptional could be omitted as is redundant to the action of transcription factors

307-313: MC and their transdifferentiated counterparts (…) lymphoma progression; should provide additional scientific detail of the study, not clear if is patients data or based on experimental models

322: colorectal cancer (CRC), already defined in line 189

329: phenotype able to revert to a full epithelial state once implanted on the subserosal connective tissue; should provide additional scientific detail of the study, not clear if is patients data or based on experimental models

330: Ex vivo and in vitro molecular analyses generated conflicting data concerning the presence (…) prognostic significance of EMT biomarkers in primary tumors, ascites and peritoneal implants; should provide additional scientific detail of the study, not clear if is patients data or based on experimental models

335: Spheroids are in fact enriched in stemness markers, such as pluripotent stem cell transcription factors Nanog, Oct4 and (…);should provide additional scientific detail of the study, not clear if is patients data or based on experimental models

363: evidenced, in tumor buds, decreased expression; evidenced, in tumor buds, decrease expression

387: What was initially considered an aberrant expression could now be represented by as a transitioning state; represented as a

394: pancreatic neoplastic cells by localizing in the intercellular junctions and interfering with E-cadherin membrane localization; unclear if pancreatic neoplastic cells are human primary tumor samples, human cell lines in vitro or experimental models (in vivo mouse models)

434 : Roberts and colleagues [8] (…) Desmocollin 3, encoded by DSC3. Additional bibliographic references supporting the role of DSC3 in cell biology and cancer should be added

450: The TGF-b/SMAD, consider using alternative state-of-the art review articles in this subject, as any from Massagué  J, for instance

519-525: BMP9 is (…) biomineralization; lacks bibliographic support

548: 4.5 Molecular pathways involved in heterotopic ossification; this section could be unified with the previous as molecular pathways for heterotopic ossification are already introduce in section 4.4

555-552: As TGF-b, the BMP signaling pathway (…) MAPK signaling; lacks bibliographic support

566: TGF-b/BMP pathway interacts with pathways of MAPK, Wnt, (…); elements/proteins from the

611: Abbreviations should appear by alphabetic order

Author Response

We gratefully acknowledge the reviewers’ suggestions, which have significantly improved the manuscript.

Reviewer #1

The review manuscript "Role of epithelial-mesenchymal plasticity in Pseudomyxoma Peritonei: implications for loco-regional treatments”, by Maria Luisa Calabrò et al., the authors combine both clinical and molecular information for a comprehensive overview of PMP pathophysiology and molecular profiling that could improve future PMP diagnosis and treatment, presenting some molecular/mechanistic evidence from other (related) cancer types (e.g. ovarian and colorectal) due to the lack of evidence from PMP. In the authors perspective, the correlation between EMT-associated mechanisms (including extraskeletal ossification), mucins production and PMP aggressiveness should be further elucidated in order to improve current diagnostic and loco-regional treatments.

This review presents a defined focus, in a topic with relevance for cancer research and therapy that lacks substantial revision in the field. The manuscript is well-written, with an eligible English, and generally well-structured and supported by relevant bibliography.

However, in some circumstances, the scientific data described cannot solely be justified by the references used (e.g. reviews) and additional/more relevant bibliography in the field should be chosen to support the mechanistic detail of cell signaling pathways (e.g TGF- TGF-b/SMAD signaling).

In the mechanistic and molecular data, authors should be more consistent and provide whenever possible, more scientific detail on the original studies, as in some circumstances is not possible to discern if the information refers to patients data or experimental models (e.g. cell lines used for in vitro studies or xenograft models, species, cancer type). To improve the scientific significance and quality of the manuscript, it is important to clearly sate if specific mechanisms were already proven to be relevant for PMP progression in patients or are still under scrutiny.

Considering the structure of the manuscript, in order to improve the clarity of presentation and interest for the reader, some revision should also be undertaken. Specifically, in some sections, there is redundant contents presented that could be combined and integrated in unified sections (e.g sections 3.1 and 3.2; sections 4.4 and 4.5 focus in related subjects) and for sake of clarity, information regarding the implication of the same molecule (e.g. Mucins) in PMP should be placed together or in subjacent sections (2.7 Mucins as therapeutic targets and and 3.3 Role of mucins in EMT and stemness).

Overall, I consider that this manuscript provides a comprehensive and significant contribution for understanding the molecular mechanisms implicated in PMP progression that could be further elucidated for improving current diagnosis and treatment, and therefore I consider that it merits publication after minor revisions.

A list of specific concerns and (minor) revisions is provided bellow.

General concerns:

Be more consistent. In some parts of the text is unclear if the data provided or conclusions withdrawn are based on experimental models or patients data (e.g “(…) as they were shown to contribute to Ovarian Cancer (OC) metastasis by secreting fibronectin [75], and to support the progression of Primary Effusion Lymphoma (PEL), (…)” as opposed to other situations were a clear and complete description is provided (“Another mucin found in PMP, MUC5B, was shown to be involved in the increased proliferation and invasiveness of breast cancer MCF7 cells in vitro, and promoted tumor growth and metastases in a xenograft model”).

If possible, be more concise in introductory sections (e.g EMT 3.1) as are supported by review articles as bibliography and part of the information is repeat in subsequent (more subject-oriented) sections (e.g. 3.2.1).

As suggested by the reviewer, subsections 2.2 and 2.7, 3.1 and 3.2 and 4.4 and 4.5 were unified, and redundancies were deleted. Subsections 2.2 + 2.7 are now subsection 2.5. The introduction has been shortened.

Specific minor revisions:

65: multi-varigated

To be more concise and to avoid redundancies, the sentence including this term has been removed. 

75: mesenchymal-epithelial transition, already defined before (62)

This oversight has been amended in Introduction (line 63).

122: types of mucin (MUC), define first time term is used (line 110)

MUC is now defined at first use (line 42).

165: AJCC, no definition in text

This abbreviation is now defined in the text (now lines 130-131).

173 according the histological grade, to the

This oversight has been amended (now line 139).

176: Table 1. Classification of PMP (modified from Carr [44]), this reference appears twice (also in the footer *modified from Carr et al. [44]).

The reference in the Title has been deleted.

Define terms/abbreviations (PMP, DPAM, PMCA, PMCA-S) in the footer.

PMP was defined in the title, whereas DPAM, PMCA and PMCA-S are now placed after their definition inside the Table (and removed from the column).

199: defective mismatch repair, dMMR; DNA mismatch repair (MMR) is more commonly used

We have readjusted the sentence with the recommended term (now line 162).

222: Although the recent introduction of immunotherapy has shown promising results in CRC patients, lacks reference

The reference has been added (now line 263).

225: 2.7 Mucins as therapeutic targets, for a better integration of concepts, and to avoid repetition of concepts, consider combining or placing in closer proximity to the 2.2 Mucins in PMP

As reported above, sections 2.2 and 2.7 have been unified in section 2.5, and shortened by deleting all redundancies.  

249: of the MAPK pathway with specific targeted drugs (i.e. MEK inhibitors); unnecessary

We deleted the unnecessary example (now line 219).

255: PMP alter the structure and the enzymatic activity of the GTPase of the a-subunit, unclear, and the enzymatic activity of the GTPase a-subunit (?)

This sentence has been corrected: the altered activity involves the GTPase domain of the alpha-subunit. (now line 225)

257: In vitro studies; lack detail for the studies referred

The functional analysis of mutant GNAS has been described (now lines 228-230).

265: The two drugs showed in vitro and in vivo in a xenograft model a synergistic effect; lacks scientific detail (cells, animal model)

This part has been integrated with additional data on the experimental models (now lines 272-276).

269: the role of a combination drug therapy with; should reformulate 

This sentence has been shortened and “combination drug therapy” was deleted (line 279).

274: 3.1 Type 3 EMT and hybrid states: a short introduction, for a better integration of concepts, and to avoid repetition, consider using this introductory info as the first paragraph of section 3.2 Role of EMP in peritoneal dissemination

This section was rearranged according to the reviewer as follows:

Subsection 3.1 “Type 3 EMT and hybrid states” was used as the introduction to subsection 3.2 (now 3.1) (dealing with type 3 EMT and hybrid states) now entitled “Current evidence of EMP and hybrid states in peritoneal metastases”, whereas the role of mesothelial cells in tumor progression (dealing with type 2 EMT) is now discussed in subsection 3.2.

289: network of factors and pathways, including a relevant group of transcription factors (Zeb1/2, Snail1/2 and Twist1/2), in addition to post-transcriptional, post-translational and epigenetic regulatory; be more concise; post-transcriptional could be omitted as is redundant to the action of transcription factors

This sentence has been shortened and modified: …. a complex network of factors and pathways, including

transcription factors (ZEB1/2, SNAIL1/2 and TWIST1/2) and other complex regulatory levels, activated by……(now lines 300-302)

307-313: MC and their transdifferentiated counterparts (…) lymphoma progression; should provide additional scientific detail of the study, not clear if is patients data or based on experimental models

Additional scientific details of this study have been provided (now lines 371-375)

322: colorectal cancer (CRC), already defined in line 189

This oversight has been amended (now defined in line 153).

329: phenotype able to revert to a full epithelial state once implanted on the subserosal connective tissue; should provide additional scientific detail of the study, not clear if is patients data or based on experimental models

Additional detailed data have been provided, with new references (now lines 316-320).

330: Ex vivo and in vitro molecular analyses generated conflicting data concerning the presence (…) prognostic significance of EMT biomarkers in primary tumors, ascites and peritoneal implants; should provide additional scientific detail of the study, not clear if is patients data or based on experimental models

Ex vivo and in vitro molecular analyses refer to studies in clinical samples  (primary tumors, ascites and metastases) as well as to studies on OC cell lines. Additional scientific details of the study have been provided (now lines 320-322).

335: Spheroids are in fact enriched in stemness markers, such as pluripotent stem cell transcription factors Nanog, Oct4 and (…);should provide additional scientific detail of the study, not clear if is patients data or based on experimental models

The experimental model is now described in detail (now lines 326-327).

363: evidenced, in tumor buds, decreased expression; evidenced, in tumor buds, decrease expression

This sentence has been reformulated: …..showed a decrease in the expression (now lines 354-355).

387: What was initially considered an aberrant expression could now be represented by as a transitioning state; represented as a

This oversight has been amended (now lines 397-398).

394: pancreatic neoplastic cells by localizing in the intercellular junctions and interfering with E-cadherin membrane localization; unclear if pancreatic neoplastic cells are human primary tumor samples, human cell lines in vitro or experimental models (in vivo mouse models)

The experimental model and confirmatory data on clinical samples have been explained (now lines 402-406).

434 : Roberts and colleagues [8] (…) Desmocollin 3, encoded by DSC3. Additional bibliographic references supporting the role of DSC3 in cell biology and cancer should be added

Additional scientific detail and bibliographic references have been added (now lines 448-455).

450: The TGF-b/SMAD, consider using alternative state-of-the art review articles in this subject, as any from Massagué  J, for instance

The reference has been replaced with one from the suggested expert in the field (now ref. 118, line 469).

519-525: BMP9 is (…) biomineralization; lacks bibliographic support

Specific references have been added (now lines 534-539, ref 135-137).

548: 4.5 Molecular pathways involved in heterotopic ossification; this section could be unified with the previous as molecular pathways for heterotopic ossification are already introduce in section 4.4

As suggested, the two sections have been unified.

555-552: As TGF-b, the BMP signaling pathway (…) MAPK signaling; lacks bibliographic support

References have been added (now lines 565-568, ref. 140-141).

566: TGF-b/BMP pathway interacts with pathways of MAPK, Wnt, (…); elements/proteins from the

As suggested, this sentence has been amended (now lines 579-580).

611: Abbreviations should appear by alphabetic order

Abbreviations have been reorganized in alphabetic order.

Reviewer 2 Report

Work of Calabro et al. aimed at EMT and its effect in Pseudomyxoma peritonei is thorough and informative. The reviewed manuscript does not generally lack on any information related to the discussed problematics and also the way it is assembled does not inspire any major criticisms.

Thus the only relatively aspects deserving revision and possible improvement are:

  1. Language presentation where it is strongly recommended to proofread the text and correct many misspelling and syntax errors which are (it is believed) due to often complex compositions of sentences.
  2. Also, in abstract the full meaning of MET should be first quoted (similar to EMT) before the actual abbreviation is used.
  3. Figures 1 and 2 - internal scales implementation are recommended
  4. What is AICC?
  5. Figure 2 - Graphical part should be improved; since the legend mentions signet-ring morphology in disseminated PMP cells, it should also be clear to readers in the graphical part of the figure (the yellow structure in the cytoplasm of one of the cells does not bring such notion). The best would be to provide real micrographs of both clustered and disseminated (single) PMP cells, also the ones with signet ring morphology.
  6. Authors should be consistent in their use of the abbreviations or genes/proteins - genes capital italics letters, proteins - capital letters. In this manuscript all variants are used inconsistently.

Author Response

Specific comments to the Reviewers

We gratefully acknowledge the reviewers’ suggestions, which have significantly improved the manuscript.

Reviewer #2

Work of Calabro et al. aimed at EMT and its effect in Pseudomyxoma peritonei is thorough and informative. The reviewed manuscript does not generally lack on any information related to the discussed problematics and also the way it is assembled does not inspire any major criticisms.

Thus the only relatively aspects deserving revision and possible improvement are:

  1. Language presentation where it is strongly recommended to proofread the text and correct many misspelling and syntax errors which are (it is believed) due to often complex compositions of sentences.

We hope to have corrected all misspellings and errors. Many long sentences have been rearranged into shorter sentences.

  1. Also, in abstract the full meaning of MET should be first quoted (similar to EMT) before the actual abbreviation is used.

The full meaning of MET has been reported in the abstract  before the abbreviation.

  1. Figures 1 and 2 - internal scales implementation are recommended

As recommended, Figures 1a and 1b have been accordingly modified, and internal scale is reported.

  1. What is AJCC?

We apologize for this oversight.  American Joint Committee on Cancer (AJCC) is now reported in extenso  when first mentioned (now lines 130-131).

  1. Figure 2 - Graphical part should be improved; since the legend mentions signet-ring morphology in disseminated PMP cells, it should also be clear to readers in the graphical part of the figure (the yellow structure in the cytoplasm of one of the cells does not bring such notion). The best would be to provide real micrographs of both clustered and disseminated (single) PMP cells, also the ones with signet ring morphology.

As requested, we have improved Figure 2 (now Figure 3) by indicating the signet-ring cell with an arrow.  We agree that original micrographs of single cells, clusters and signet-ring cells might add value to the Figure. However, the primary aim of this Figure is to schematically summarize the switch in the EMT state between single and clustered cells in PMP, therefore we consider that the histological features are not necessary.

  1. Authors should be consistent in their use of the abbreviations or genes/proteins - genes capital italics letters, proteins - capital letters. In this manuscript all variants are used inconsistently.

We have double-checked the nomenclature to be as consistent as possible.

Reviewer 3 Report

This manuscript contains well-organized information but needs to improve.

1) Title means there are some parts of loco-regional treatment.

--> But It isn't easy to find them.

2) section about Pseudomyxoma peritonei

--> please add some figure for explaining the contents of it. 

3) In all parts, they want to say EMP but there are only EMT but not EMP.

4) Please clarify the MUC2 and EMT. Does it induce or inhibit EMT?

5) Please add a figure about the role of mucins in cancer and PMP.

6) Molecular profile --> of what?

7) In some Figure, please indicate the meaning of colour in cells.

8) Please emphasize the role of ossification in EMT, if possible. Does blocking of ossification revert EMT?

9) CK20?

10) Please add or emphasize some prospects more clearly.

Author Response

Specific comments to the Reviewers

We gratefully acknowledge the reviewers’ suggestions, which have significantly improved the manuscript.

Reviewer #3

This manuscript contains well-organized information but needs to improve.

1) Title means there are some parts of loco-regional treatment.

--> But It isn't easy to find them.

As suggested, we have now expanded subsection 2.6 on “Current therapies” by also including details on PIPAC treatment.

2) section about Pseudomyxoma peritonei

--> please add some figure for explaining the contents of it. 

As suggested by the reviewer, we have now added a new Figure (now Figure 2) illustrating the steps of PMP pathogenesis.

3) In all parts, they want to say EMP but there are only EMT but not EMP.

In this review, we have used EMP when data indicated different hybrid, transitioning states. The use of EMT or MET is related to the direction of the transition, whereas EMP is related to the flexibility of the whole process. We have now replaced  some “EMT” with “EMP” according to this criterion for consistency. 

4) Please clarify the MUC2 and EMT. Does it induce or inhibit EMT?

MUC2 is involved in maintenance of the epithelial state in colon cancer, and thus possibly plays a role in EMT suppression. In accordance with this,  MUC2 inhibition augments the aggressiveness of colon cancer cell lines in terms of in vitro migratory capabilities and in vivo metastatic potential, as reported in the review (Section 3.3, lines 386-390). Concerning PMP, no data are presently available on a role of MUC2 in the EMT state of PMP cells. As we discussed in the review (Section 4.3, lines 502-508), PMP cells produce large amounts of MUC2 but they do not maintain a consistent, fully epithelial profile. While peritoneal implants and clusters in ascites usually show more epithelial markers, scattered cells in mucin pools (usually denoting more aggressive tumors) have a more mesenchymal phenotype, suggesting more complex and context-dependent mechanisms of EMT modulation in this pathologic condition. This is now specified in the text (lines 507-508).

5) Please add a figure about the role of mucins in cancer and PMP.

As the main aim of this manuscript is to review and discuss the role of epithelial-mesenchymal plasticity in PMP pathogenesis, we think that the Figures should focus on this topic; the role of mucins in cancers has been extensively depicted in reviews dealing with this topic, which are among our references. The role of mucins in the pathogenesis of PMP is now shown in Figure 2.

6) Molecular profile --> of what?

The title of subsection 2.4 has been changed: “Molecular profile of PMP”

7) In some Figure, please indicate the meaning of colour in cells.

We have now indicated the signet-ring cell with an arrow in Figure 3 (ex Figure 2) and mentioned it in the legend. 

8) Please emphasize the role of ossification in EMT, if possible. Does blocking of ossification revert EMT?

Ossification is a very interesting but also extremely rare phenomenon in PMP. We think we have quite extensively discussed the possible pathogenetic implications and mechanisms of this process (as stated, it is found in very few PMP cases) in subsection 4.4 and Figure 4 (subsections 4.4 and 4.5 were unified).  We have also briefly reported on the hypothetic role of this process in the generation of cells adapted to a bone microenvironment and thus possibly involved in bone metastases (lines 526-529). Therefore, it is conceivable that the selective inhibition of extraskeletal ossification might be achieved through blockade of specific factors, such as BMP9, acting like EMT-inducers, and might have important effects in one of the most common metastatic sites.      

9) CK20?

This oversight has been amended (now line 506).

10) Please add or emphasize some prospects more clearly.

As requested, we have now extended the last section.

Round 2

Reviewer 3 Report

All issues from me were cleared.